



# Halogen activation and radical cycling initiated by imidazole-2-carboxaldehyde photochemistry

Pablo Corral Arroyo[1, 2], Raffael Aellig[3], Peter A. Alpert[1], Rainer Volkamer[4, 5], Markus Ammann[1,*]

[1]Paul Scherrer Institute, Laboratory of Environmental Chemistry, 5232 Villigen PSI, Switzerland.
[2]Department of Chemistry and Biochemistry, University of Bern, 2012 Bern, Switzerland.
[3]ETH Swiss Federal Institute of Technology Zürich, Institute for Atmospheric and Climate Science, 8006 Zurich, Switzerland.
[4]Department of Chemistry and Biochemistry, 215 UCB, University of Colorado, Boulder, CO 80309, USA
[5]Cooperative Institute for Research in Environmental Sciences (CIRES), 216 UCB, University of Colorado,
Boulder, CO 80309, USA

*Correspondence to*: Markus Ammann (markus.ammann@psi.ch)

**Abstract.** Atmospheric aerosol particles can contain light absorbing organic compounds, also referred to as brown carbon (BrC). In the context of the ocean surface and of sea spray aerosol deriving from the latter, light absorbing organic species are
also referred to as chromophoric dissolved organic matter (CDOM). Many BrC or CDOM species (especially carbonyls, dicarbonyls or aromatic carbonyls such as imidazole-2-carboxaldehyde (IC)), referred to as photosensitizers, form triplet excited states upon UV-VIS light absorption. These triplet excited states are strong oxidants and may initiate catalytic radical reaction cycles within atmospheric aerosol particles and at their surface, therefore increasing the reactive oxygen species (ROS) production within atmospheric aerosol particles. Triplet states (or ROS resulting from them) can also react with halides
generating halogen radicals and additionally molecular halogens compounds, which can be released into the gas phase and may thus contribute to halogen activation.  In this work we study the influence of bromide and iodide on the photosensitized $HO_2$ production and release upon UV irradiation of films in a coated wall flow tube (CWFT) containing IC in a matrix of citric acid (CA). Additionally we measured the iodine release upon irradiation of IC/CA films in the CWFT. We use a kinetic model to interpret our results and to assess radical production and iodine release in sea-spray particles. As indicated by the
experimental results and confirmed by the model, significant recycling of halogen species occurs via scavenging reactions with $HO_2$, to prevent the full and immediate release of the molecular halogen (bromine and iodine) produced, while partially shutting down the $HO_x$ chemistry. The recycling efficiency is higher and affected by diffusion limitations at high viscosity. Our findings also show that halides can increase substantially the BrC or CDOM photosensitized $HO_2$ production (which in turn promotes radical and ROS production) by reacting with triplet statesin sea-spray particles. The iodine production within
sea salt aerosol particles due to iodide oxidation by ozone is estimated at $5.9 \times 10^{-5}$ M s$^{-1}$ assuming ozone equilibration in the particle. Under diffusion limitation this activation can drop several orders of magnitude in an aged, organic-rich sea-spray derived aerosol ($1.1 \times 10^{-7}$ M s$^{-1}$ for an ozone diffusion coefficient of $10^{-12}$ cm$^2$ s$^{-1}$). The estimated iodine production from BrC photochemistry amounts to $2.5 \times 10^{-7}$ M s$^{-1}$. This indicates that BrC photochemistry can exceed $O_3$ reactive uptake in



controlling the rates of iodine activation from sea spray particles under dry or cold conditions where diffusion is slow within particles.

# 1 Introduction

Volatile halogen-containing species such as $CH_3X$, $CH_2XY$, HOX, XY, and $X_2$ (where X and Y can be Cl, Br and I), also called activated halogen species (AHS), are produced at the ocean surface, in snowpacks or in aerosol particles and emitted into the atmospheric gas phase. The production of AHS is driven by oxidation of halides by ozone (Carpenter et al., 2013) (Schmidt et al., 2016) or by radicals (OH or $NO_3$ for example) (Sander and Crutzen, 1996), or photochemical oxidation (Wang and Pratt, 2017; Wren et al., 2013). They can also reach the atmosphere by biogenic emissions of halogen-containing organic species (Org-X) (Hepach et al., 2016; Vogt et al., 1999), or by emissions from volcanos, among other processes (Simpson et al., 2015). Halogen activation refers to the production of activated halogen species. These species are direct precursors of reactive halogen species (RHS) such as X atom or XO (Sherwen et al., 2016a), which trigger oxidative processes in the gas phase (Saiz-Lopez et al., 2012). In the troposphere, the presence of RHS shifts the $HO_x$ equilibrium ($HO_2 \leftrightarrow OH$) towards OH (Bloss et al., 2005; Chameides and Davis, 1980; Lary, 1996; Saiz-Lopez, 2012; Sommariva et al., 2012; von Glasow et al., 2004), especially for the case of IO (Bloss et al., 2005; Dix et al., 2013; Saiz-Lopez et al., 2008; Schmidt et al., 2016; Stone et al., 2018; Volkamer et al., 2015). RHS also influence the budgets of nitrogen oxides ($NO_x$), organic compounds and organic peroxy radicals (Simpson et al., 2015). It has been observed that RHS of iodine produce ultrafine particles found in coastal aereas (Mahajan et al., 2011; McFiggans et al., 2010). This new particle formation occurs via polymerization of $I_2O_5$ (Hoffmann et al., 2001; McFiggans et al., 2004; Saunders and Plane, 2006; Sherwen et al., 2016b; Sipila et al., 2016), which is produced by the (photo)oxidation of iodine precursor species such as $I_2$ (Saiz-Lopez and Plane, 2004), HOI (Carpenter et al., 2013; Sherwen et al., 2016b) and Org-X (Carpenter, 2003). The production and cycling of activated halogen species at the ocean surface or in sea-spray particles are key processes to understand their release into the gas phase and the contributions to their emission fluxes (Carpenter et al., 2013; Herrmann et al., 2003; Pechtl et al., 2007).

Photochemistry can trigger many oxidative processes in the atmosphere which contribute directly to the oxidative budget both in the condensed and gas phases of the atmosphere, by producing oxidizing excited molecular states and radicals such as triplet states, singlet oxygen or $HO_x$ radicals (Canonica, 2000; George et al., 2015). Brown carbon (BrC) is defined as the fraction of organic compounds in atmospheric aerosol particles that absorbs efficiently in the UVA-VIS range. Some BrC species can undergo direct photolysis, while others may also be photosensitizers, which are species that photocatalyze radical chain reactions, involving organic and $HO_x$ radicals, via excited triplet states as well established in aquatic photochemistry (Canonica, 2000). BrC species may be primary organic compounds (e.g., from biomass burning) or result from multiphase oxidation processes (Laskin et al., 2015). A wealth of light absorbing species also occurs at the ocean surface or in terrestrial water, in both cases originating from the degradation of biological material, which are referred to as chromophoric dissolved



organic matter (CDOM) (Chen et al., 2016; Rochelle-Newall and Fisher, 2002). Recently, photosensitizing BrC species have also recently been recognized as contributors to the oxidant budget in airborne particles (Corral-Arroyo, 2018; George et al., 2015; González Palacios et al., 2016; Yu et al., 2014). Oxidation of halides by the triplet states of photosensitizers (Tinel, 2014) or by chlorophyll (Reeser et al., 2009), thus CDOM proxies, precedes halide radical chemistry at the sea water surface

(Jammoul et al., 2009), and likely also in sea spray aerosol particles, and contributes to halogen activation. Even though quantification of the triplet forming species in aerosol particles is still in its infancy (Kaur and Anastasio, 2018), the concentration of organic matter and potential chromophores as precursors for triplets are significant in marine aerosol particles (Blanchard, 1964; Chen et al., 2016; Cincinelli et al., 2001; Hoffman and Duce, 1976; Hunter and Liss, 1977; O'Dowd and de Leeuw, 2007). From the concentration of triplet states in fog water of up to $10^{-13}$ M (Kaur and Anastasio, 2018), the upper limit

of the concentration of triplet states in aerosol particles would be around $10^{-10}$ M due to concentration at low water content. The concentration of halides in sea spay aerosol particles may reach $10^{-6}$ M for iodide (Baker, 2004, 2005; Pechtl et al., 2007) and $8 \times 10^{-3}$ M for bromide (Herrmann et al., 2003). Assuming a rate coefficient of the reaction between a typical sensitizer triplet state and iodide of $5 \times 10^{9}$ M$^{-1}$ s$^{-1}$ (Tinel, 2014), iodine activation may reach $2.5 \times 10^{-7}$ M s$^{-1}$, which leads to a low life time of iodide in the aqueous phase. This indicates that photosensitized chemistry may contribute significantly to the halogen

activation in sea-spray particles.

Typical photosensitizers of interest are carbonyls, which absorb above 300 nm especially when attached to an aromatic system (see absorption spectra in SI Figure S1) (Canonica, 2000). Figure 1 illustrates the catalytic cycle of a photosensitizer in an organic aerosol particle in presence of halides. First, the photosensitizer (P) absorbs radiation, and this excitation step is followed by singlet (P*(s)) to triplet (P*(t)) intersystem crossing. The triplet state is long lived and, thus, can act as an oxidant

(Canonica, 2000) reacting with an electron donor, such as a halide ion (X$^-$), or an organic H atom donor, producing a ketyl radical (PH$^\bullet$/P$^{\bullet-}$). Oxygen competes with electron/H atom donors for the triplet being able to produce singlet oxygen ($^1$O$_2$) from its reaction with the triplet. The ketyl radical passes on an electron or hydrogen atom to oxygen or another electron acceptor (e.g., NO$_2$ (Stemmler et al., 2006)) producing HO$_2$. The photosensitizer catalytic cycle is enclosed in box a). The efficiency of the catalytic cycle is reduced by deactivation of the singlet, deactivation of the triplet (phosphorescence, non-radiative decay

and reaction with oxygen) and other radical reactions involving the reduced ketyl radical. The presence of organics that are highly reactive with triplet states increases the photosensitized HO$_2$ radical production of imidazole-2-carboxaldehyde (IC) up to 20 M day$^{-1}$ (Corral-Arroyo, 2018). Subsequent to the oxidation of the halide anion by the triplet state of IC, halide radicals (X$^\bullet$ and X$_2^-$) are produced and the ensuing halide radical-radical reactions produce molecular halogen compounds (Reactions 8-11 and 14, Table 1). H$_2$O$_2$ is additionally produced by HO$_2$ itself reaction and by the reaction between HO$_2$ and X$_2^-$. We do

not consider further reactivity of H$_2$O$_2$ since it is not photolyzed at our wavelengths. The oxidized species X$_2$, X$_2^-$ and X$^\bullet$ are likely recycled into X$^-$ by HO$_2$ radicals (Reactions 5-9, Table 1), however a fraction of X$_2$ may be released into the gas phase (Jammoul et al., 2009), and these recycling processes are determining the effective efficiency of halogen activation. De Laurentiis and co-workers suggested that excited triplet states may oxidize bromide faster than OH radicals in sea water (De Laurentiis et al., 2012). Some modelling studies of aerosol chemistry consider halogen chemistry driven entirely by inorganic





halogen chemistry (Sherwen et al., 2016a; Sherwen et al., 2016b), while Pechtl et al. claimed that dissolved organic matter may be included as a HOI deactivation pathway (Pechtl et al., 2007; Sarwar et al., 2016). The contribution of photosensitized halogen activation is missing in these models.

Imidazole-2-carboxyldehyde (IC) is a BrC proxy (absorption spectrum in Fig. S1) and well-known photosensitizer (Corral-Arroyo, 2018; González Palacios et al., 2016). Imidazoles, which include IC, are BrC compounds formed as products from the multi-phase chemistry of glyoxal and ammonium sulfate (AS) in aqueous aerosols (Aregahegn et al., 2013; Kampf et al., 2012; Yu et al., 2014). Glyoxal is an important oxygenated volatile organic compound (OVOC) and its dominant known sources on the global scale are biogenic VOC (Stavrakou et al., 2009). Citric acid (CA) serves as a proxy for non-absorbing highly oxidized and functionalized secondary organic compounds in the atmosphere, which are also ubiquitous in marine air (O'Dowd and de Leeuw, 2007). In solution, CA takes up or releases water gradually without phase change over the whole range of relative humidity (RH) values studied here (Lienhard et al., 2012; Zardini et al., 2008).

In this work we quantify the effect of bromide and iodide on the $HO_2$ production from IC photochemistry and evaluate the iodine activation resulting from the subsequent condensed phase radical reactions by means of Coated Wall Flow Tube (CWFT) experiments. We measured the iodine and $HO_2$ release from films loaded with IC, CA and bromide or iodide while irradiating with UV light. Finally, we discuss the relevance of our findings for atmospheric sea spray aerosol.

## 2 Experimental

### 2.1 Experimental description

The setup to indirectly detect $HO_2$ production in an irradiated laminar coated wall flow tube (CWFT) by scavenging $HO_2$ with an excess of nitrogen monoxide (NO) has been described in detail in our previous work (Corral-Arroyo, 2018; González Palacios et al., 2016) and in the SI (Fig S2 and S3). Tubes (1.2 cm inner diameter, 50 cm long, Duran glass) coated with mixtures of IC/CA/NaX were snuggly fitting into the temperature and relative humidity controlled CWFT as inserts surrounded by 7 fluorescent lamps (UV-A range, Philips Cleo Effect 20W: 300–420 nm, 41 cm, 2.6 cm o.d., see SI Fig. S1). The flows of $N_2$ and $O_2$ were set at 1 L min⁻¹ and 0.5 L min⁻¹ respectively. The NO concentration (5-10 ml min⁻¹ of a mix of $N_2$ and NO at 100ppm) was always high enough ($1 - 2.5 \times 10^{13}$ molecules per cm³) to efficiently scavenge ~99% of $HO_2$ produced by the films within 20-50 ms and thus far less than our residence time of 2 s. NO was measured by a chemiluminescence detector (Ecophysics CLD 77 AM). For experiments with bromide we can assume that the concentration of bromide did not change over the time scale of our experiments and, therefore, the system was in steady-state under irradiation. On the other hand, the concentration of iodide decreased rapidly (within tens of minutes), since the iodine is rapidly released into the gas phase, so we assessed the NO loss from the first few minutes of irradiation for reporting $HO_2$ production rates.

Iodine release into the gas phase was observed by converting all gas phase iodine compounds to $I_2O_5$ following a procedure developed by Saunders et al. (Saunders and Plane, 2006). Part of the flow from the reactor (0.1L min⁻¹ out of 1.5L min⁻¹) was mixed with 0.2 L min⁻¹ of $O_2/O_3$ (1%), and this mixture was fed into a quartz reactor with 0.07 s residence time, which is



irradiated with a Hg penray lamp (184 nm). The $O_2/O_3$ (1%) mixture was produced by a discharge in pure $O_2$ and quantified with a photometric ozone analyzer. All iodine compounds are readily photolyzed and oxidized to $I_2O_5$, which polymerized and produced particles via homogeneous nucleation (Carpenter et al., 2013; Saunders and Plane, 2006). The flow was led to a Scanning Mobility Particle Analyzer (SMPS) though aerosol tubing with a residence time of around 20 seconds and the mass

of the $I_2O_5$ particles was determined from measuring their size distribution with the SMPS consisting of a home-made differential mobility analyzer (DMA, 93.5 cm long, 0.937 cm inner diameter 1.961 outer diam.) and a Condensation Particle Counter (CPC, Model 3775). The $I_2O_5$ particle density was assumed to be 2.3±0.3 g cm$^{-3}$ following Saunders et al. (Saunders and Plane, 2006). We were able to measure particles reliably only ≥20 nm in diameter (Fig. S4, SI). This method does not distinguish between iodine and any other volatile iodine compound, which can be oxidized up to $I_2O_5$. HOI or IO might be

produced in the films by oxidation of halide radicals or molecular halogen. We rely on our proposed mechanism (Figure 1) and assume that iodine activation is dominated by production of $I_2$.

Aqueous solutions containing halides ($10^{-8}$ M, $10^{-5}$ M and 0.01 M for iodide and $10^{-5}$ M and 0.01 M for bromide) were prepared beforehand. For each experiment, 76.6 mg of CA and 4 mg of IC (2.5 mg of IC for the experiments measuring iodine release) were dissolved in different volumes of a halide solution in order to get different halide concentrations in the films. This solution

was deposited in the glass tube while rolling and turning the tube in all directions at room temperature under a gentle flow of $N_2$ humidified to the RH later used in experiments. This procedure is necessary to ensure homogeneous thin films checked by visual inspection. Freshly prepared solutions were used to prepare the films. Concentrations in the film were 6 M for CA, 0.7 M for IC, between $10^{-8}$ M and 0.01 M for iodide and between $10^{-4}$ and 0.01 M for bromide (0.4 M of IC and 33mM of iodide for iodine release measurements) at around 35% RH at 20°C. These were calculated assuming that the water content in the

film was controlled by the hygroscopicity of CA only, as parameterized by Zardini et al. (Zardini et al., 2008).

## 2.2 Chemicals

The chemicals used were imidazole-2-carboxaldehyde (>99%, Aldrich), citric acid (Fluka), sodium bromide (Sigma-Aldrich) and sodium iodide (Sigma-Aldrich).

## 3 Results

### 3.1 HO₂ production, scavenging and release

Figure 2 presents the HO$_2$ radical release in the CWFT as a function of halide concentration from films loaded with 4 mg of IC and 76.6 mg of CA and 0.7-70 µg of sodium bromide or $10^{-3}$-300 µg of sodium iodide which equate to $10^{-4}$-$10^{-2}$ M of bromide and $3 \times 10^{-7} - 3 \times 10^{-2}$ M of iodide. Error bars are the standard deviation of several measurements in the same film. For iodide just two measurements were made for each film, since iodide is consumed rapidly, while for bromide we made 4-6

measurements for each film. One measurement is made by the comparison of the signal of NO before and after switching on or off the UV lamps. Experiments employed constant IC and CA concentration at 0.7 M and 6 M, respectively. For comparison,



HO$_2$ production for the IC/CA system without any halides is shown as the blue solid line in Fig. 3, consistent with our previous studies (Corral-Arroyo, 2018; González Palacios et al., 2016). The photosensitized oxidation of CA resulting in the production of the ketyl radical (PH$^\bullet$, see Figure 1), followed by reaction of PH$^\bullet$ with O$_2$ to lead to the formation of HO$_2$ (upper red arrow in Figure 1), in absence of halide ions, has been discussed in detail in those studies.

Once the halide concentration is increased, it starts to contribute to the reduction of P* due to its ability to donate an electron. This leads to increased production of PH$^\bullet$ and thus increased production of HO$_2$ (Figure 1). The observed HO$_2$ production and release is enhanced from $10^{-7}$ M of iodide and $10^{-4}$ M for bromide, which implies a faster rate coefficient for the reduction of the IC triplet (P*) by iodide than that for reduction by bromide. Tinel et al. (Tinel, 2014) measured the rate coefficients between the triplet state of IC and bromide and iodide (reaction 5) as $5.33 \times 10^9$ M$^{-1}$ s$^{-1}$ and $6.27 \times 10^6$ M$^{-1}$ s$^{-1}$, respectively. The
difference is roughly three orders of magnitude. This is in agreement with our results since the ratio of the minimum concentrations at which iodide and bromide provide faster ketyl radical production and thus faster HO$_2$ release than CA alone ($10^{-7}$ M for iodide and $10^{-4}$ M for bromide) is comparable to the ratio of these two rate coefficients.

After the oxidation of the halide ion by the triplet state, a cascade of fast reactions takes place leading to the production of X$_2^-$ and molecular halogens (X$_2$). Most of these species, including the molecular halogen compounds, react rapidly with HO$_2$
(reactions 5-9 in Table 1) leading to the drop of the HO$_2$ release at high concentrations of halides (higher concentrations of halides induce higher concentrations of halide radicals). Additionally HO$_2$ radicals also react with each other meaning that this scavenging pathway will be more relevant at high concentrations of halides, where more HO$_2$ is produced ($8 \times 10^5$ M$^{-1}$ s$^{-1}$) (Bielski et al., 1985).

The HO$_2$ scavenging reactions shown in Table 1 are faster for iodide species than for bromide species, which induces a
suppression of the HO$_2$ release at lower concentrations for iodide than for bromide. In this way, HO$_2$ is mostly scavenged before being released into the gas phase for films with concentrations of iodide above $10^{-3}$ M and of bromide of $10^{-2}$ M. The ratio of the rate coefficients of the triplet with iodide and bromide (R5) is higher than the ratio of the rate coefficients of HO$_2$ with iodide and bromide species which induce the recycling (R12-16). We suspect that this is the reason why the HO$_2$ release drops faster with concentration for bromide than for iodide.

**Table 1.** Chemical reactions and the corresponding rate coefficients of halide and HO$_2$ radical chemistry

| No | Reaction | Rate coefficient (X=Br) M$^{-1}$ s$^{-1}$ | Rate coefficient (X=I) M$^{-1}$ s$^{-1}$ | Reference |
|----|----------|-------------------------------------------|------------------------------------------|-----------|
| R1 | IC → IC$^{3*}$ | $1 \cdot 10^{-3}$* | $1 \cdot 10^{-3}$* | Corral-Arroyo |
| R2 | IC$^{3*}$ + O$_2$ → IC + $^1$O$_2$ | $3 \cdot 10^9$ | $2.6 \cdot 10^9$ | Canonica |
| R3 | IC$^{3*}$ → IC | $6.5 \cdot 10^5$* | $6.5 \cdot 10^5$* | Corral-Arroyo |
| R4 | IC$^{3*}$ + CA → ICH$^\bullet$ + CA$^\bullet$ | 90 | 90 | Corral-Arroyo |
| R5 | IC$^{3*}$ + X$^-$ → IC$^{\bullet-}$ + X$^\bullet$ | $6.27 \cdot 10^6$ | $5.33 \cdot 10^9$ | Tinel |





| R6 | $ICH^{\bullet} + O_2 \rightarrow IC + HO_2^{\bullet}$ | $1 \cdot 10^9$ | $1\text{-}5 \cdot 10^9$ | Maillard |
|---|---|---|---|---|
| R7 | $HO_2^{\bullet} + HO_2^{\bullet} \rightarrow H_2O_2$ | $8 \cdot 10^5$ | $8.3 \cdot 10^5$ | Bielski |
| R8 | $X^- + X^{\bullet} \rightarrow X_2^{-\bullet}$ | $9 \cdot 10^9$ | $1.1 \cdot 10^{10}$ | Nagarajan/Ishigure |
| R9 | $X_2^- + X^{\bullet} \rightarrow X_3^-$ | - | $8.4 \cdot 10^9$ | Ishigure |
| R10 | $X^{\bullet} + X^{\bullet} \rightarrow X_2$ | - | $1.9 \cdot 10^{10}$ | Ishigure |
| R11 | $X_2 + X^- \leftrightarrow X_3^-$ | $2.7 \cdot 10^{4\ E}$ | $768^E$ | Bianchini/Morrison |
| R12 | $HO_2^{\bullet} + X^{\bullet} \rightarrow O_2 + HX$ | $1.6 \cdot 10^8$ | - | Wagner |
| R13 | $HO_2^{\bullet} + X_2^{-\bullet} \rightarrow O_2 + HX + X^-$ | $1 \cdot 10^8$ | - | Wagner |
| R14 | $HO_2^{\bullet} + X_2^{-\bullet} \rightarrow HO_2^- + X_2$ | $9.1 \cdot 10^7$ | $4 \cdot 10^9$ | Wagner/Ishigure |
| R15 | $HO_2^{\bullet} + X_2 \rightarrow O_2 + X_2^{-\bullet}$ | $1.5 \cdot 10^8$ | $1.8 \cdot 10^7$ | Bielski/Schwarz |
| R16 | $HO_2^{\bullet} + X_3^- \rightarrow X^- + H^+ + O_2 + X_2^{-\bullet}$ | $<1 \cdot 10^7$ | - | Bielski |
| R17 | $X_2 \overset{h\nu}{\rightarrow} 2\,X^{\bullet}$ | - | $0.01*$ | -/Choi |

Source of rate coefficients: (Bianchini and Chiappe, 1992; Bielski et al., 1985; Canonica, 2000; Choi et al., 2012; Corral-Arroyo, 2018; Ishigure et al., 1988; Maillard et al., 1983; Morrison et al., 1971; Nagarajan and Fessenden, 1985; Schwarz and Bielski, 1986; Tinel, 2014; Wagner and Strehlow, 1987) *First order rate coefficient ($s^{-1}$). $^E$Equilibrium constant ($M^{-1}$).

A kinetic model was developed about IC photochemistry assuming steady state in our recent work (Corral-Arroyo, 2018), where we estimated the $HO_2$ release from films of IC/CA as a function of concentration of IC, relative humidity, film thickness or additional triplet scavengers. For the present case, we adapted that model, now including the scavenging of the triplet state of IC by halides (Tinel, 2014) (reaction 4) (instead of an additional organic donor) and the inter-halogen conversion reactions (reactions 8-11) as well as the set of $HO_2$ scavenging reactions 12 – 16 (Table 1). We added the photolysis of iodine by
integrating the irradiance spectrum of the lamps used (Fig. S1) and the absorption spectrum of iodine (Choi et al., 2012). We treated $I_2$ and $I_3^-$ as the same species. Further details of the reactions and rate coefficients are given in the SI. We found that the $HO_2$ release was underpredicted at middle and high concentrations of halides. We decided to keep the inter-halogen conversion reactions (reactions 8-11) at their literature values and tune the $HO_2$ scavenging reactions 12 – 16. To obtain reasonable model results, they were reduced as explained in the SI. The model gets the general trend with a maximum and a
slope down upon increasing concentrations of halides with a shift towards higher concentrations. The model results allow us to assess the $HO_2$ release (Figure 2), and the efficiency in the cycling of the radicals, which will be explained further below. As apparent from Figure 2, important differences between observation and best model output remain. These differences might come from different rate coefficients of scavenging of triplet states by halides and of the $HO_2$ scavenging reactions in dilute aqueous solutions (where the used rate coefficients were measured) versus those in the high solute strength solution of the
present study and prevalent in atmospheric aerosol. There is evidence that hydrogen bonded transition states are involved in electron transfer (IvkovicJensen and Kostic, 1997), proton coupled electron transfer, hydrogen abstraction reactions (Mitroka





et al., 2010) and quenching reactions between triplets and salts (Kunze et al., 1997). However, we refrained from adding more and ill-constrained processes and parameters to achieve better apparent fit. The position of the maximum is determined by the ratio between the scavenging of triplet states by halides and the $HO_2$ scavenging reactions. Since we kept the rate coefficient of the scavenging of the triplets fixed, the tunable parameters were the $HO_2$ scavenging reactions. We observed that our model

was overpredicting the $HO_2$ radical release at middle and high concentrations of halides ($10^{-5} – 10^{-1}$ M). Therefore, to obtain reasonable model results we reduced the values of the rate coefficients of scavenging of $HO_2$ by halide radicals (R11-15) keeping them equaled to each other (SI). In spite of the differences, the model correctly predicts the stronger slope of the decrease of $HO_2$ release with halide concentration for the case of bromide in comparison to that for iodide.

### 3.2 Iodine activation

We performed very similar CWFT experiments in which the iodine release was measured as described in the experimental section. The CWFT was loaded with 2.5 mg of IC, 76.6 mg of CA (6.5% in molar ratio) and 313 µg of NaI, corresponding to concentrations of 0.4M, 6M and 33mM of IC, CA and iodide respectively, and the iodine release into the gas phase at 34% RH was followed uninterruptedly. The $HO_2$ release was measured separately with a separate film under the same conditions and within the same range of time.

Figure 3 shows the release of iodine obtained from the measurements of $I_2O_5$ particles by the SMPS versus time. The particle mass concentration was obtained from the mobility diameter by assuming a density of 2.3 g cm$^{-3}$ and spherical particles. This particle mass was converted to an equivalent $I_2$ release assuming the stoichiometry of $I_2O_5$. The profile of the release shows a peak after the first ten minutes of irradiation and decays over the following 60 minutes until the release ceases. Figure 3 also presents the corresponding $HO_2$ release versus time, which is entirely depleted in the beginning, as expected for the high iodide

concentration (see Figure 2), and then increases linearly until 90 minutes before it reaches a steady state at $3 \times 10^{11}$ molecules min$^{-1}$ cm$^{-2}$, which is the same as that measured in absence of iodide (blue arrow in Figure 3). When comparing to Figure 2, the evolution of the $HO_2$ release with time indicates that most likely a drop in the iodide concentration from 33mM to below $10^{-4}$ M occurs. The maximum in the iodine release was observed after several minutes of irradiation being about $5.5 \times 10^{13}$ molecules min$^{-1}$ cm$^{-2}$. The iodine release model prediction is $4.9 \times 10^{13}$ molecules min$^{-1}$ cm$^{-2}$ at the initial concentration of

iodide. In view of the uncertainty of many of the parameters used in the model, the consistency between model and observation is rather surprising. Photolysis and scavenging rate coefficients can be different in aqueous solutions (where the used rate coefficients were measured) and in organic matrix (IvkovicJensen and Kostic, 1997; Kunze et al., 1997; Lignell et al., 2014; Mitroka et al., 2010), and also the simple steady state assumption is bearing uncertainties. It seems that iodine release is a more robust observable than HO2 radicals, and the model apparently captures the halogen chemistry reasonably well. The total

integrated $I_2O_5$ mass measured over the whole observation period corresponds to 70(±10) % of the iodide added to the film. The synchronized behavior of both releases ($HO_2$ and $I_2$) indicates that iodide is significantly consumed after 100 minutes of irradiation and presumably most of iodide is converted into molecular iodine. As indicated in the SI, we could not measure the mass from particles smaller than 20 nm of diameter, so the mass calculated is a lower limit of the real mass released. An



alternative possible sink of halides in the films is the reaction of halide radicals ($I^{\bullet}$ or $I_2^-$) and of HOI or HOBr with organics producing Org-X (Abrahamsson et al., 2018; Gilbert et al., 1988) or further oxidation of iodine to iodate, which was beyond the scope of our study.

The efficiency of the iodine activation depends on the different competing processes occurring in the P catalytic cycle and the ones involving halogen radical chemistry (Figure 1). Oxygen, CA and halides compete for the triplet. Once the triplet oxidizes the halide, the radicals produced can be recycled back to halide (recycling A) or produce the molecular $X_2$ compounds bromine and iodine. These can be recycled back to $X_2^-$ (recycling B) or escape to the gas phase. For iodine, the model predicts that around 50% of halogen atoms produced are released to the gas phase as molecular halogen (40% RH, $D_{HO2} = 3.5 \times 10^{-12}$ cm$^2$ s$^{-1}$ and $D_{I2} = 2 \times 10^{12}$ cm$^2$ s$^{-1}$) indicating that the fate of around half of iodide radicals is the recycling and the other half is leaving the condensed phase as iodine. This is the overall result of several competing chemical processes. $HO_2$, $X^{\bullet}$ and $X^-$ are competing for $X_2^-$, and finally, after $X_2$ production, $X_2$ can diffuse out or react with $HO_2$ to produce $X_2^-$. Our model predicts that the ~ 50 % of the I radicals ($I^{\bullet}$ and $I_2^-$) produced are recycled back to iodide (around 40% for bromide) and these numbers do not change significantly with RH. Based on the model, the efficiency in the release of molecular iodine or bromine once they are produced is then about 85 – 95 % and > 99 % respectively (5 – 15 % and < 1 % recycling back to $X_2^-$ respectively). Upon decreasing the diffusion coefficient by one order of magnitude, the efficiency in the release of molecular iodine or bromine, once they are produced, is then about 45 – 65 % and 97.5 % respectively (15 – 50 % and 2,5 % recycling back to $X_2^-$, respectively). On the other hand, upon increasing the diffusion coefficient by one orders of magnitude the efficiency in the release of molecular iodine or bromine once they are produced is then about 97 − 99.5 % for iodine and almost 100% for bromine.

## 4 Conclusions and atmospheric implications

In this work we show the influence of halides on the photochemistry of imidazole-2-carboxaldehyde and its oxidative capacity. Both iodide and bromide can increase significantly the $HO_2$ radical production in the system IC/CA since the oxidation of halide ions by IC triplet is several orders of magnitude faster than the corresponding oxidation of CA (when $[I^-] > 10^{-6}$ M then $k_I \cdot [I^-] > k_{CA}[CA]$) (Corral-Arroyo, 2018; Tinel, 2014). At the same time, the halogen radical species resulting from the reaction with the triplet scavenge away the $HO_2$ produced preventing it to leave the film and thus maintaining the capacity to red-ox cycle with the halide species.

The concentration of halides in sea spay aerosol particles go up to $10^{-6}$ M for iodide (Baker, 2004, 2005; Pechtl et al., 2007) and $8 \times 10^{-3}$ M for bromide (Herrmann et al., 2003). At the sea surface many kinds of chromophoric organic compounds are present, including biomolecules, carbonylic and carboxylic compounds (CDOM) (Chen et al., 2016; Quinn et al., 2015), which are uplifted together with sea spray particles (Cincinelli et al., 2001; Hunter and Liss, 1977). Based on our results, halides are concentrated enough in atmospheric aerosol particles to contribute to the radical production. CA is likely a reasonable proxy for oxidized secondary organic compounds present in aerosol particles after some aging time. Primary organics present in





nascent sea spray particles or on the ocean surface may themselves scavenge triplet states with faster rates than CA and in the same order of magnitude as iodide (Canonica, 2000), thus diminishing the capacity for halogen activation.

First we calculate the iodine produced internally by reaction between triplets and iodide. Assuming a concentration of $10^{-10}$ M for triplet states and $10^{-6}$ M for iodide in sea-spray particles and a second order rate coefficient of $5 \times 10^{-7}$ M$^{-1}$ s$^{-1}$ (Tinel, 2014),

the first order iodine activation by photosensitized chemistry may get to $5 \times 10^{-7}$ M s$^{-1}$. We compare this rate with iodine activation from the oxidation of iodide by ozone. At 25°C, the mean thermal velocity of ozone is 318 m s$^{-1}$. Assuming ozone Henry's law constant (H$_{O3}$) of 0.14 M atm$^{-1}$ (Berkemeier et al., 2016), a diffusion coefficient of ozone (D$_{O3}$) of $1 \times 10^{-12}$ cm$^2$ s$^{-1}$ in a viscous organic particle (Berkemeier et al., 2016), which corresponds to an aqueous CA particle at ~ 40% RH at room temperature or to a CA particle at ~70% RH at -20°C (Lienhard et al., 2014), and $k_{O3/I^-}$ as $4.2 \times 10^9$ M$^{-1}$ s$^{-1}$ (Magi et al., 1997)

the ozone uptake coefficient would be $2.7 \times 10^{-8}$ (eq.1) under the assumption that the reaction proceeds in the reacto-diffusive regime.

$$\gamma = \frac{4H_{O_3}RT}{\omega_{O_3}}\sqrt{D_X k_b^{II}[I]_b} \quad (1)$$

Where $H_{O_3}$ is the Henry's law constant, $R$ is the gas constant, $T$ is temperature, $\omega_{O_3}$ is the mean thermal velocity of ozone, $D_x$ is the diffusion coefficient, $k_b^{II}$ is the rate coefficient of the reaction between ozone and iodide ($k_{O3/I^-}$ as $4.2 \times 10^9$ M$^{-1}$ s$^{-1}$) and

15 $[I]_b$ is the concentration of iodide in aerosol particles. Note that equation (1) does not fully capture all details of the reaction kinetics, as on one hand the kinetics may switch from this reaction-diffusion limited regime to a bulk reaction limited regime, depending on particle size, diffusivity of O$_3$ and iodide concentration. In addition, the contribution of a surface reaction remains open (see Moreno et al. (Moreno et al., 2018) for an in depth analysis of this kinetics).

For a gas-phase concentration of 100 ppb and a particle 500 nm in diameter, the uptake of ozone ($U$) (eq. 2) and potential iodine activation will be $1.1 \times 10^{-7}$ M s$^{-1}$.

$$U = 2\pi C_{g,O_3}\omega_{O_3}r^2\gamma \quad (2)$$

Where $C_{g,O_3}$ is the concentration of ozone in the gas phase and $r$ is the radius of the particle.

On the other hand, when assuming that ozone is fully equilibrated with the aqueous phase of the particle (assuming $H_{O_3} = 0.14$

M atm$^{-1}$), thus at high RH, where diffusive mixing of the aqueous phase is fast, at atmospheric concentrations (100ppb) the iodine activation will be $5.9 \times 10^{-5}$ M s$^{-1}$.

We conclude that photosensitized iodine production is relevant for aerosol sea spray particles containing chromophores under dry conditions when the reactive uptake of ozone is slow. Under humid conditions the activation via reaction with ozone will likely be more relevant. We noted the existence of a cycling in halide radical chemistry that shuts down the HO$_x$ chemistry

and, simultaneously, prevent the release of molecular halogens to the gas phase. This cycling strongly depends on the diffusion



properties of the matrix, reaching a greater cycling efficiency when diffusion is low and lower efficiency when diffusion is fast. Even so, the release is not completely depleted under a wide range of diffusion regimes and a large fraction (50%-100%) will be released. Based on the model predictions, we suspect that the same processes are occurring for bromide. De Laurentiis and co-workers suggested that excited triplet states oxidize faster bromide than OH radicals in sea water (De Laurentiis et al.,

2012). This conclusion, together with this work, highlights the role of CDOM or BrC species in halide chemistry in sea spray aerosol particles.

*Code and data availability.* The data for simulations performed under Sect. 3.1 and 3.2 are available in…

*Author contributions.* The scientific contributions were provided by all coauthors.

*Competing interests.* The authors declare that they have no conflict of interest.

*Acknowledgements.* We would like to thank Mario Birrer for technical support. M.A., P.A., and P.C.A. appreciate support by the Swiss National Science Foundation (Grant 163074). P.A. thanks for funding from the European Union's Horizon 2020 research and innovation program under the Marie Skłodowska-Curie grant agreement No 701647. R.V. thanks for the financial support from NSF-AGS-1620530.

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



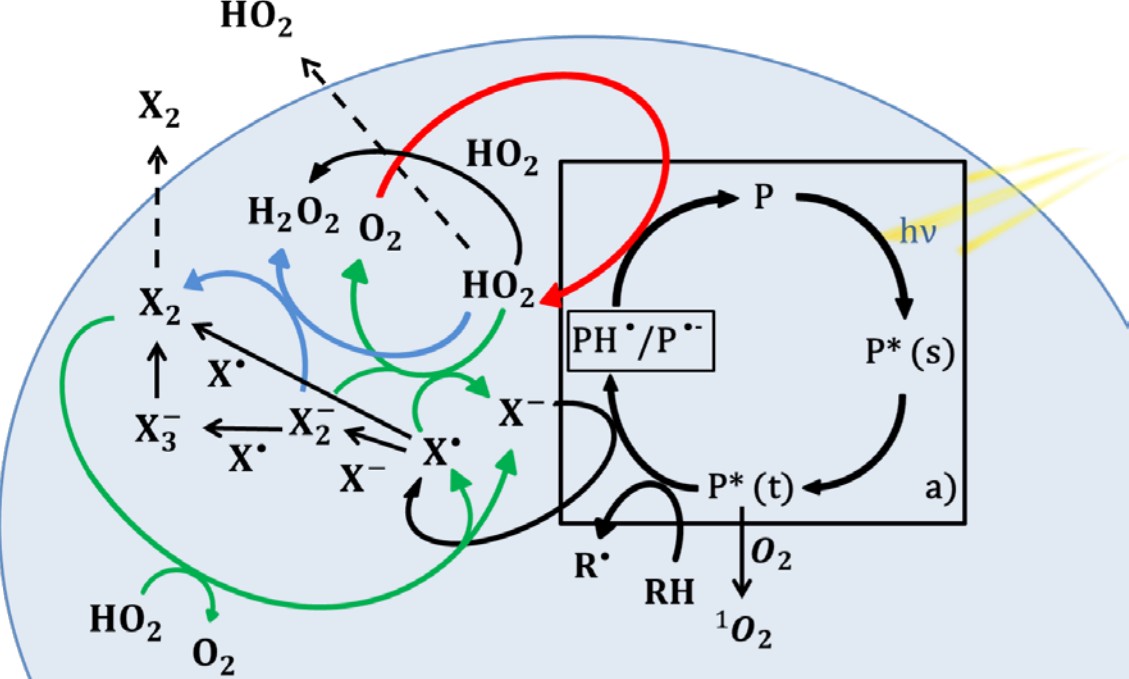

**Figure 1. Photochemical catalytic cycle of IC (box a) and halide radical chemistry induced in a particle. IC first absorbs light leading to the triplet state, which reacts with an H atom/electron donor (DH and X⁻) to produce the reduced ketyl radical (PH·) and halide radicals (X·). The halide radicals can produce molecular halogen ($X_2$) or $X_2^-$ by reacting with X⁻. PH· may transfer an H atom or electron to an acceptor, such as $O_2$ producing $HO_2$ radicals. $HO_2$ can recycle the halide radicals previously produced into halides or oxidize further the $X_2^-$ to produce halogen molecules. $HO_2$ radicals can be released into the gas phase or react within the particle with halide radicals or with itself. Solid lines refer to reactions and dashed lines refer to transfer from condensed to gas phase. Red reaction arrows indicate reactions promoting $HO_2$ production, green arrows indicate reactions of recycling of halides promoted by $HO_2$ and blue arrows indicate reactions of $X_2$ promoted by $HO_2$. Rate coefficients are provided in Table 1.**



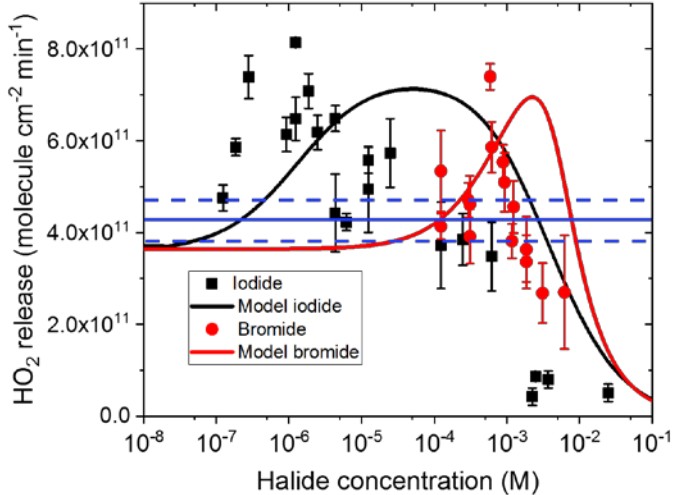

**Figure 2. HO₂ release at 34% RH from films with 4 mg of IC, 76.8 mg of CA and various concentrations of bromide (red circles) and iodide (black squares). Error bars indicate the standard deviation of between 2-6 measurements in the same film. The blue line and dashed blue lines indicate measured HO₂ production and uncertainty, respectively, from films with the same IC and CA concentration but in absence of halides. Solid lines are fits using the model described in the text below.**





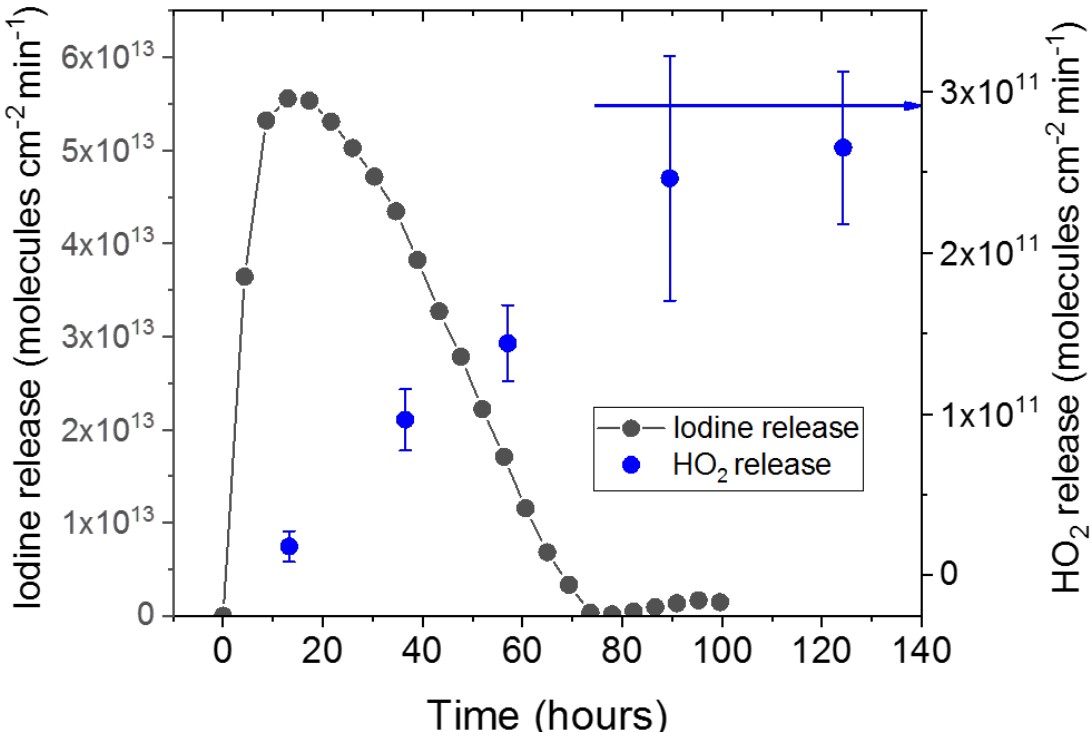

**Figure 3. Iodine release (black), measured as iodine oxide particles produced by oxidation of iodine species released from the CWFT expressed as equivalent $I_2$ release (left y-axis), and $HO_2$ release (blue, right y-axis) into the gas phase versus time while irradiating a film in the CWFT loaded with 2.5 mg of IC, 76.8 mg of CA and 313 μg of NaI and equilibrated at 34% RH. The blue arrow indicates the $HO_2$ release expected for the film in absence of iodide.**

**Table 1.** Chemical reactions and the corresponding rate coefficients of halide and $HO_2$ radical chemistry

| No | Reaction | Rate coefficient (X=Br) $M^{-1}s^{-1}$ | Rate coefficient (X=I) $M^{-1}s^{-1}$ | Reference |
|---|---|---|---|---|
| R1 | $IC \rightarrow IC^{3*}$ | $1 \cdot 10^{-3*}$ | $1 \cdot 10^{-3*}$ | Corral-Arroyo |
| R2 | $IC^{3*} + O_2 \rightarrow IC + {}^1O_2$ | $3 \cdot 10^9$ | $2.6 \cdot 10^9$ | Canonica |
| R3 | $IC^{3*} \rightarrow IC$ | $6.5 \cdot 10^{5*}$ | $6.5 \cdot 10^{5*}$ | Corral-Arroyo |
| R4 | $IC^{3*} + CA \rightarrow ICH^{\bullet} + CA^{\bullet}$ | $90$ | $90$ | Corral-Arroyo |
| R5 | $IC^{3*} + X^- \rightarrow IC^{\bullet -} + X^{\bullet}$ | $6.27 \cdot 10^6$ | $5.33 \cdot 10^9$ | Tinel |
| R6 | $ICH^{\bullet} + O_2 \rightarrow IC + HO_2^{\bullet}$ | $1 \cdot 10^9$ | $1-5 \cdot 10^9$ | Maillard |
| R7 | $HO_2^{\bullet} + HO_2^{\bullet} \rightarrow H_2O_2$ | $8 \cdot 10^5$ | $8.3 \cdot 10^5$ | Bielski |



| R8 | $X^- + X^{\bullet} \to X_2^{-\bullet}$ | $9 \cdot 10^9$ | $1.1 \cdot 10^{10}$ | Nagarajan/Ishigure |
|---|---|---|---|---|
| R9 | $X_2^- + X^{\bullet} \to X_3^-$ | - | $8.4 \cdot 10^9$ | Ishigure |
| R10 | $X^{\bullet} + X^{\bullet} \to X_2$ | - | $1.9 \cdot 10^{10}$ | Ishigure |
| R11 | $X_2 + X^- \leftrightarrow X_3^-$ | $2.7 \cdot 10^{4\,E}$ | $768^E$ | Bianchini/Morrison |
| R12 | $HO_2^{\bullet} + X^{\bullet} \to O_2 + HX$ | $1.6 \cdot 10^8$ | - | Wagner |
| R13 | $HO_2^{\bullet} + X_2^{-\bullet} \to O_2 + HX + X^-$ | $1 \cdot 10^8$ | - | Wagner |
| R14 | $HO_2^{\bullet} + X_2^{-\bullet} \to HO_2^- + X_2$ | $9.1 \cdot 10^7$ | $4 \cdot 10^9$ | Wagner/Ishigure |
| R15 | $HO_2^{\bullet} + X_2 \to O_2 + X_2^{-\bullet}$ | $1.5 \cdot 10^8$ | $1.8 \cdot 10^7$ | Bielski/Schwarz |
| R16 | $HO_2^{\bullet} + X_3^- \to X^- + H^+ + O_2 + X_2^{-\bullet}$ | $<1 \cdot 10^7$ | - | Bielski |
| R17 | $X_2 \xrightarrow{h\nu} 2\,X^{\bullet}$ | - | $0.01^*$ | -/Choi |

Source of rate coefficients: (Bianchini and Chiappe, 1992; Bielski et al., 1985; Canonica, 2000; Choi et al., 2012; Corral-Arroyo, 2018; Ishigure et al., 1988; Maillard et al., 1983; Morrison et al., 1971; Nagarajan and Fessenden, 1985; Schwarz and Bielski, 1986; Tinel, 2014; Wagner and Strehlow, 1987) *First order rate coefficient ($s^{-1}$). $^E$Equilibrium constant ($M^{-1}$).