# Peer review of "Halogen activation and radical cycling initiated by imidazole-2-carboxaldehyde photochemistry"

_Atmospheric Chemistry and Physics, 2019_

## Referee Comment (RC1) · Anonymous Referee #1 · 28 Mar 2019

This paper present for first time a new mechanism for halogens to be emitted that have not been considered up to date and that in determined conditions could be even more important that O3 uptake in see surface. The paper is well written and structured and only require that authors address the following technical comments:

- Please comment about the importance of differences between the spectra lamps and radiation in the atmosphere and possible implications for this study and atmospheric implications. - Some comments should be added related the possible effect of Cl- in the mechanism proposed since in sea salt aerosols Cl- is going to be present. - This sentence need to be finished up: "Code and data availability. The data for simulations

performed under Sect. 3.1 and 3.2 are available in..."

---

## Referee Comment (RC2) · Anonymous Referee #2 · 16 Apr 2019

This is a nice contribution aiming at understanding how triplet states chemistry may induce both halogen activation and HO2 release. Imidazole-2-carboxaldehyde (IC) was used a proxy for chromophoric dissolved organic matter (CDOM) or brown carbon (BrC) in coated wall flow tube experiments. Its chemistry was simulated by a simple "box" model, which was adjusted to the measured yields. The experiments and calculations are performed according to the current best standards.

While the manuscript itself is well written, it could nevertheless benefit from some reediting as some sentences are repeating between the experimental and result sections.

This manuscript is definitively suitable for publication in Atmospheric Chemistry and

[Figure]

Physics, and I would raise only a very few minor comments.

While the films were prepared from aqueous solutions, it unclear from the experimental section if those stay liquid during the experiments or if they were drying out. The authors should make clear in which phase the experiments were performed. There is only a few superficial mention about the relative humidity set during the experiments, which may affect both the phase and temperature of the films.

In order to adjust to the measurements, the authors decided to keep the inter-halogen conversion reactions (reactions 8-11) at their literature values and tune the HO2 scavenging reactions 12 – 16 (or 11-15 as stated elsewhere in the text). To obtain reasonable model results, they were reduced. Here as I wondering if the authors have thoughts on the possible influence of the CA. radicals produced in reaction R4? Very recently, Roveretto et al (ACS Earth Space Chem., 2019, 3 (3), pp 329–334) reported, in similar experiments, between those organic and inorganic radicals. While this is not affecting the conclusions made here, it might explain the need of adjusting part of the rate constants in Table 1.

---

## Author Comment (AC1) · 2 Jul 2019

Below, we provide a point-by-point response to the comments raised by reviewer 1

Reviewer: This paper present for first time a new mechanism for halogens to be emitted that have not been considered up to date and that in determined conditions could be even more important that O3 uptake in see surface. The paper is well written and structured and only require that authors address the following technical comments: - Please comment about the importance of differences between the spectra lamps and radiation in the atmosphere and possible implications for this study and atmospheric implications.

[Figure]

Response: Thank you for this comment. We will include the following text in the conclusion of the revised version: "At 0âŮę zenith angle the ratio of the excitation rates of IC by sunlight (spectrum given in Figure S1) and that by the lamps used in our experiments (jsun/jlamps) is about 2.83. Therefore, in terms of irradiation our results can be extrapolated to the atmosphere with confidence."

Reviewer: Some comments should be added related the possible effect of Cl- in the mechanism proposed since in sea salt aerosols Cl- is going to be present.

Response: Thank you this comment. We will include the following text in the conclusion of the revised version: "Assessment of chlorine activation via IC as chromophore and sensitizer reacting with chloride, which is present in higher concentrations in sea salt aerosol particles (∼5.4 M) (Herrmann et al., 2003), was beyond the scope of this study. While the ratio of chloride to bromide or iodide is higher than the inverse ratio of the corresponding rate coefficients (Tinel et al., 2014), the complex radical chemistry and kinetics requires detailed attention to understand impacts on chlorine activation and photosensitized HO2 production."

Reviewer: This sentence need to be finished up: "Code and data availability. The data for simulations...

Response: Yes, of course, we will deposit the data underlying each figure in the supporting material, and adapt this statement accordingly in the revised version.

References

Herrmann, H.: Kinetics of Aqueous Phase Reactions Relevant for Atmospheric Chemistry, Chem. Rev., 103, 4691-4716, 10.1021/cr020658q, 2003.

Tinel, L. D., Stéphane George, Christian: A time-resolved study of the multiphase chemistry of excited carbonyls: Imidazole-2-carboxaldehyde and halides, C. R. Chimie, 17, 801-807, 2014.

---

## Author Comment (AC2) · 2 Jul 2019

Below, we provide a point-by-point response to the comments raised by reviewer 2.

Reviewer: This is a nice contribution aiming at understanding how triplet states chemistry may induce both halogen activation and HO2 release. Imidazole-2-carboxaldehyde (IC) was used a proxy for chromophoric dissolved organic matter (CDOM) or brown carbon (BrC) in coated wall flow tube experiments. Its chemistry was simulated by a simple "box" model, which was adjusted to the measured yields. The experiments and calculations are performed according to the current best standards. While the manuscript itself is well written, it could nevertheless benefit from

some reediting as some sentences are repeating between the experimental and result sections. This manuscript is definitely suitable for publication in Atmospheric Chemistry and Physics, and I would raise only a very few minor comments.

Response: We would like to thank the reviewer for this positive assessment. We agree with the comment regarding the language, and accordingly we will provide a thorough check of the language and consider some streamlining of parts of the text and remove redundancy among the different sections in the revised version.

Reviewer: While the films were prepared from aqueous solutions, it unclear from the experimental section if those stay liquid during the experiments or if they were drying out. The authors should make clear in which phase the experiments were performed. There is only a few superficial mention about the relative humidity set during the experiments, which may affect both the phase and temperature of the films.

Response: Comment well taken: While we have characterized citric acid coatings in the cited previous studies in more detail, this has not been clearly enough described here. Therefore, we will include the following sentence in the experimental description: "Since films were never dried below 35 % RH, they are expected to remain concentrated liquid aqueous solutions at 35 % RH with a viscosity around 10 - 100 Pa s."

Reviewer: In order to adjust to the measurements, the authors decided to keep the inter-halogen conversion reactions (reactions 8-11) at their literature values and tune the HO2 scavenging reactions 12 – 16 (or 11-15 as stated elsewhere in the text). To obtain reasonable model results, they were reduced. Here as I wondering if the authors have thoughts on the possible influence of the CA. radicals produced in reaction R4? Very recently, Roveretto et al (ACS Earth Space Chem., 2019, 3 (3), pp 329–334) reported, in similar experiments, between those organic and inorganic radicals. While this is not affecting the conclusions made here, it might explain the need of adjusting part of the rate constants in Table 1.

Response: Agreed. We will include the following sentence in the results section: "CA

daughter radicals can react with halide radicals and produce halogen-containing organic compounds, as already observed in aquatic media (Roveretto et al., 2019). This can result in a partial scavenging of the halide radicals and it might be an explanation for the need to decrease the rate coefficients for R11 to R15."

References

Roveretto, M., Li, M., Hayeck, N., Brüggemann, M., Emmelin, C., Perrier, S., and George, C.: Real-Time Detection of Gas-Phase Organohalogens from Aqueous Photochemistry Using Orbitrap Mass Spectrometry, ACS Earth and Space Chemistry, 3, 329-334, 10.1021/acsearthspacechem.8b00209, 2019.

---

## Author Response (AR1)

Dear editor

In response to the review comments, we have first implemented the changes as announced in our response to the review comments in ACPD. Then, we have substantially improved the narrative of the text, text flow and structure and English grammar, which required quite some moving around within the individual sections, but no change to the overall structure. A few more minor changes in the presentation of the implications have been done, which are described after the actions resulting from the review comments. At the end of this document, the manuscript in track changes mode is appended.

Below we repeat the comments (plain), our response (bold) and the actual action in the revised version (italics),

Reviewer 1:

- Please comment about the importance of differences between the spectra lamps and radiation in the atmosphere and possible implications for this study and atmospheric implications.

**Thank you for this comment. We will include the following text in the conclusion:**
**"At $0°$ zenith angle the ratio of the excitation rates of IC by sunlight (spectrum given in Figure S1) and that by the lamps used in our experiments ($j_{sun}/j_{lamps}$) is about 2.83. Therefore, in terms of irradiation our results can be extrapolated to the atmosphere with confidence."**

*Final action in text: We have added the following sentence to the conclusions on line 11 of page 11: "We note that at 0° zenith angle, the solar actinic flux is about 3 times greater than the UV lamps we used in the experiment, and thus excitation rates of IC may be 3 times faster than what was used here."*

- Some comments should be added related the possible effect of Cl- in the mechanism proposed since in sea salt aerosols Cl- is going to be present.

**Thank you this comment. We will include the following text in the conclusion:**

**"Assessment of chlorine activation via IC as chromophore and sensitizer reacting with chloride, which is present in higher concentrations in sea salt aerosol particles (~5.4 M) (Herrmann et al., 2003), was beyond the scope of this study. While the ratio of chloride to bromide or iodide is higher than the inverse ratio of the corresponding rate coefficients (Tinel et al., 2014), the complex radical chemistry and kinetics requires detailed attention to understand impacts on chlorine activation and photosensitized HO$_2$ production."**

*Final action in text: We have added the following sentence to the conclusions at the bottom/top of pages 9/10: "Assessment of chlorine activation via IC as chromophore and sensitizer reacting with chloride, which is present in higher concentrations in sea salt aerosol particles (~5.4 M) (Herrmann et al., 2003), was beyond the scope of this study. While the ratio of chloride to bromide or iodide is higher than the inverse ratio of the corresponding rate coefficients (Tinel et al., 2014), the complex radical chemistry and kinetics require detailed attention to understand impacts on chlorine activation and photosensitized HO$_2$ production."*

- This sentence need to be finished up: "Code and data availability. The data for simulations

**Yes, of course, we will deposit the data underlying each figure in the supporting material, and adapt this statement accordingly in the revised version.**

*Final action in text: The code and data availability statement now reads: "The data underlying Figures 2 and 3 and the matlab codes of the steady-state model calculations are available as supporting files."*

Reviewer 2

…The experiments and calculations are performed according to the current best standards. While the manuscript itself is well written, it could nevertheless benefit from some reediting as some sentences are repeating between the experimental and result sections. This manuscript is definitely suitable for publication in Atmospheric Chemistry and Physics, and I would raise only a very few minor comments.

**We agree with the comment regarding the language, and accordingly we will provide a thorough check of the language and consider some streamlining of parts of the text and remove redundancy among the different sections.**

*Final action in text: As mentioned above, we have substantially edited the whole manuscript again to improve text flow, conciseness and English language. A few specific important changes on top of that will be mentioned at the end of this description of revisions.*

While the films were prepared from aqueous solutions, it unclear from the experimental section if those stay liquid during the experiments or if they were drying out. The authors should make clear in which phase the experiments were performed. There is only a few superficial mention about the relative humidity set during the experiments, which may affect both the phase and temperature of the films.

**We agree with this comment. While we have characterized citric acid coatings in the cited previous studies in more detail, this has not been clearly enough described here. Therefore, we will include the following sentence in the experimental description:**

**"Since films were never dried below 35 % RH, they are expected to remain concentrated liquid aqueous solutions at 35 % RH with a viscosity around 10 - 100 Pa s."**

*Final action in text: "Once prepared, a solution was deposited in the glass tube while rolling and turning the tube in all directions at room temperature under a gentle flow of $N_2$ humidified to the RH later used in experiments. This procedure was necessary to ensure homogeneous thin films checked by visual inspection and to prevent the film from drying out prior to the experiments…. Films are expected to be liquid at 35 % RH and have a viscosity of 10 - 100 Pa s (Song et al., 2016)."*

In order to adjust to the measurements, the authors decided to keep the inter-halogen conversion reactions (reactions 8-11) at their literature values and tune the HO2 scavenging reactions 12 – 16 (or 11-15 as stated elsewhere in the text). To obtain reasonable model results, they were reduced. Here as I wondering if the authors have thoughts on the possible influence of the CA. radicals produced in reaction R4? Very recently, Roveretto et al (ACS Earth Space Chem., 2019, 3 (3), pp 329–

334) reported, in similar experiments, between those organic and inorganic radicals. While this is not affecting the conclusions made here, it might explain the need of adjusting part of the rate constants in Table 1.

**Agreed. We will include the following sentence in the results section:**

**"CA daughter radicals can react with halide radicals and produce halogen-containing organic compounds, as already observed in aquatic media (Roveretto et al., 2019). This can result in a partial scavenging of the halide radicals and it might be an explanation for the need to decrease the rate coefficients for R11 to R15)."**

*Final action in text: We have added a citation to Roveretto et al. (2019) in the introduction section on page 3, line 25, where such processes are mentioned at the general level. In section 3.1, page 8, line 13, we have added: "This can be explained if CA derived radicals reacted with halogen radicals to produce halogen-containing organic compounds, as already observed in aquatic media (Roveretto et al., 2019), which could result in a partial scavenging of halogens. " Then, in the iodine activation section 3.2, on page  9, line 1, we have added:  "Alternatively, sinks of halides in the films could be the reaction of halide radicals ($I^\bullet$ or $I_2^-$) and of HOI or HOBr with organics producing Org-X (Abrahamsson et al., 2018; Gilbert et al., 1988; Roveretto et al., 2019) or further oxidation of iodine to iodate, which was beyond the scope of our study."*

*Further changes to the manuscript*

*The discussion of the model application on page 7 has been improved to better represent the differences between model output and measurements.*

*We noticed an error in reporting the model prediction for iodine release for the initial film conditions, which is in fact much higher than what was actually measured, on page 7, line 27. This is reasonable, because the measurement cannot resolve a sharp initial release of iodine: "The maximum of the iodine release was $5.5 \times 10^{13}$ molecules $min^{-1}$ $cm^{-2}$. When normalized to the initial amount of iodide present in the film, this corresponds to a iodide life-time of around 8400 s, thus a bit more than 2 hours. The steady-state model prediction is $4.9 \times 10^{15}$ molecules $min^{-1}$ $cm^{-2}$ at the initial concentration of iodide, which cannot be directly compared to the measurement, because the measurement with the SMPS could not resolve a sharp initial release. Note in addition that the model is not following the system over time." We refrain from further speculating on model uncertainty at this point in the text, since this is discussed in the previous section.*

*In the previous manuscript version, we have only used the simple back-of-the-envelope calculation made in the introduction section about iodide oxidation by triplet excited states to compare to iodide oxidation rates induced by ozone, thus without actually using the results of the present study. We therefore added the results of a few additional model estimates based on the present study: "Based on the results obtained, we can refine this number. We note that the experiment in Figure 3 cannot be directly extrapolated to atmospheric conditions due to the high 33 mM iodide concentration used, which suppresses the triplet concentration to $10^{-12}$ M. In addition, the viscous films were 3.4 µm thick, thus beyond atmospheric particle size ranges. We therefore run model calculations with $10^{-6}$ M iodide, a film thickness of 0.5 µm and with the IC concentration adjusted such that the triplet*

*concentration at steady state reached $10^{-10}$ M. Under these conditions, the iodine release is estimated at $4.1 \times 10^{-7}$ M s$^{-1}$ at 35% RH and roughly a factor of 2 larger when the diffusion coefficients are set to the $10^{-6}$ cm$^2$ s$^{-1}$ range for a low viscosity liquid."* This allowed to sharpen the conclusion. The numbers have also been updated in the abstract.

*We corrected equation (2) by a factor 2, since it refers to $O_3$ uptake (and thus iodide, I$^-$, oxidation), and not $I_2$ release.*

[revised manuscript text omitted]